# Central Line-Associated Bloodstream Infection Due to *Elizabethkingia anophelis*: Case Report and Literature Review on Pediatric Infections

**DOI:** 10.3390/microorganisms12061145

**Published:** 2024-06-04

**Authors:** Sofia Maraki, Nikolaos Katzilakis, Ioannis Neonakis, Dimitra Stafylaki, Viktoria Eirini Mavromanolaki, Ioannis Kyriakidis, Iordanis Pelagiadis, Eftichia Stiakaki

**Affiliations:** 1Department of Clinical Microbiology and Microbial Pathogenesis, University Hospital of Heraklion, 71110 Heraklion, Greece; sofiamaraki@yahoo.gr (S.M.); ineonakis@gmail.com (I.N.); 2Department of Pediatric Hematology-Oncology, Laboratory of Blood Diseases and Childhood Cancer Biology, University Hospital of Heraklion, Medical School, University of Crete, 71003 Heraklion, Greece; kyriakidis@med.uoc.gr (I.K.); ipelagiadis@pagni.gr (I.P.); efstel@uoc.gr (E.S.); 3Department of Pediatrics, Agios Nikolaos General Hospital, 72100 Agios Nikolaos, Greece; stafylaki.dimitra@gmail.com (D.S.); sofiamaraki@gmail.com (V.E.M.)

**Keywords:** *Elizabethkingia anophelis*, pediatric infections, neonate, identification, antimicrobial susceptibility

## Abstract

*Elizabethkingia anophelis* is an opportunistic pathogen causing lifethreatening infections in humans, particularly in immunocompromised patients, neonates and the elderly. We report a case of central line-associated bloodstream infection by *E. anophelis* in a 2.5-year-old girl with acute lymphoblastic leukemia successfully treated with a combination of piperacillin/tazobactam and amikacin. The literature was also reviewed on pediatric infections caused by *E. anophelis*, focusing on clinical manifestations, underlying medical conditions, treatment and outcome. Accurate identification with MALDI-TOF, or using molecular techniques, is of the utmost importance because treatment and prognosis differ depending on the species. Considering that *E. anophelis* is multiresistant to antibiotics and that inappropriate antimicrobial therapy is an independent risk factor for mortality, the early, accurate identification of bacterial species and prompt effective treatment are essential to achieve optimal therapeutic outcomes.

## 1. Introduction

*Elizabethkingia* species are aerobic, glucose-nonfermenting, catalase-positive, oxidase-positive, and indole-positive Gram-negative bacilli widely distributed in natural environments such as soil, water, and plants, as well as in healthcare settings [1]. *Elizabethkingia*, formerly known as *Flavobacterium*, was first designated by an American microbiologist at the CDC, Elizabeth O. King, in 1959 [2]. In 1994, it was reclassified in the genus *Cryseobacterium,* and then in 2005, based on 16S rRNA gene sequencing, it was placed in the new genus *Elizabethkingia* [3,4]. Currently, the genus *Elizabethkingia* comprises six species, namely, *E. meningoseptica*, *E. anophelis*, *E. miricola*, *E. bruuniana*, *E. ursingii*, and *E. occulta* [1]. *E. anophelis* is an opportunistic pathogen most commonly affecting infants or critically ill adults with underlying comorbidities [1,5]. It is particularly known to cause neonatal sepsis and meningitis, especially in premature newborns, and sometimes is involved in outbreaks of life-threatening infections, with mortality rates ranging from 24% to 60% [1,5,6,7,8].

Herein, we describe a case of central line-associated bloodstream infection (CLABSI) due to *E. anophelis* in a 2.5-year-old girl with acute lymphoblastic leukemia and review the literature on pediatric cases caused by *E. anophelis.*

## 2. Case Description

A 2.5-year-old girl was diagnosed with acute lymphoblastic leukemia of the B lineage (B-ALL). The full blood count (FBC) at diagnosis was WBC: 2300/mm^3^, Hb: 6.9 g/dL and PLT: 16,000/mm^3^. The myelogram showed full infiltration by lymphoblasts and the immonophenotyping revealed common pre-B ALL (EGIL classification). A central venous catheter (CVC) Hickman type was inserted, and the patient was started on intensive chemotherapy according to the ALL IC-BFM 2009 protocol. Due to prognostic factors and treatment response, she was classified to receive treatment of the intermediate-risk group.

In a febrile neutropenia episode three months post starting intensive chemotherapy, *Streptococcus mitis* was isolated from blood cultures taken from the CVC. Based on the results of the susceptibility testing, the patient was given teicoplanin as a loading dose at 10 mg/kg every 12 h intravenously for three doses, followed by a maintenance dose of 10 mg/kg once daily, along with teicoplanin lock therapy. The CVC was kept in place. The patient was started again on chemotherapy according to the protocol.

Seven months after diagnosis and a month before ending the intensive protocol when the patient was receiving cytarabine 70 mg/m^2^/d and thioguanine 60 mg/kg/m^2^, she became febrile. Blood, urine, stool and pharyngeal cultures were taken, and the patient was started on empirical treatment with intravenous piperacillin/tazobactam at a dosage of 300 mg/kg every 6 h. The full blood count was WBC: 200/mm^3^ with absolute neutrophil count (ANC): 0/μL, Hb: 8.6 g/dL, PLT: 38,000/μL and CRP: 4.5 mg/dL (normal value < 0.5 mg/dL). Chemotherapy was stopped during this episode. Blood specimens taken from both the CVC and the peripheral veins on the first day of the febrile episode were inoculated into BacT/Alert PF bottles and incubated in a BacT/Alert 3D blood culture system (BioMérieux, Marcy L’Etoile, France). Both sets were positive for the same Gram-negative microorganism, *Elizabethkingia anophelis* as identified by matrix-assisted laser desorption ionization–time of flight (MALDI-TOF) mass spectrometry (VITEK MS system, BioMérieux; version 3.2). The identification was further confirmed by the DNA sequencing of the 16S ribosomial RNA gene. The alignment with the Genbank database showed 100% identity with *E. anophelis* isolate CP034594.1. The derived sequence was assigned the accession number PP579760.1 in the GenBank data library. Blood cultures obtained through the CVC became positive 210 min earlier, establishing the diagnosis of CLABSI. The blood cultures continued to be positive for *E. anophelis*, and the patient remained febrile for 3 days.

The in vitro susceptibility for selected antimicrobials was performed by the gradient strip method (E-test, BioMérieux). The antimicrobial susceptibility pattern of the isolate is presented in Table 1.

Based on the profile of the antibiogram, amikacin 20 mg/kg every 24 h was added. Amikacin was added as an adjuvant antibiotic because of the complementary action that can be expected with piperacillin/tazobactam. Four days after the beginning of the episode the patient became afebrile and blood cultures remained negative. Despite the initially very low white blood cell and platelet counts post intensive chemotherapy, clinical and biochemical improvement was observed. Intravenous antibiotics were continued for 10 days and the CVC was preserved. Currently, the patient is receiving maintenance treatment with oral chemotherapeutical agents.

The source of the infection remained undetermined because the microorganism was not isolated from any of the environmental samples (water supplies, surfaces and medical equipment).

Ethics committee name: the Ethics Committee of the University Hospital of Heraklion, Crete, Greece, Approval Code: 13030, Approval Date: 30 April 2024.

## 3. Discussion

*E. anophelis* was initially isolated from the midgut of the Anopheles gambiae mosquito in 2011 [9]. The first reported clinical case of *E. anophelis* infection was meningitis in an 8-day-old girl in the Central African Republic. *E. anophelis* was identified by 16S-rRNA sequencing [10]. Since this initial report, sporadic cases of serious systemic infections in infants and adults and several outbreaks of *E. anophelis* have been reported in Asia and the USA. The largest outbreak was registered in the Midwestern United States, resulting in 20 deaths among 65 infected patients [6]. To date, in Europe, only two adult cases and one outbreak of *E. anophelis* have been described [7,11,12]. In many previous studies, it has been revealed that the incidence of *E. anophelis* infections was highly underestimated due to *E. anophelis* being misidentified as *E. meningoseptica* based on phenotypes, prior MALDI-TOF systems not including *E. anophelis* in their diagnostic databases, and MALDI-TOF with amended databases [1,8]. MALDI-TOF, with updated databases validated for clinical application for this species, and molecular methods such as 16S rRNA sequencing and whole-genome sequencing (WGS) are reliable and accurate in species identification.

In Medline/Pubmed, searching the keywords “*Elizabethkingia anophelis* pediatric infections”, we found only 21 previously reported cases [8,10,13,14,15,16,17,18,19,20,21,22]. Our case is the first pediatric *E. anophelis* infection described in Europe. Table 2 summarizes the patient characteristics, the clinical manifestations, underlying medical conditions, the type of specimen cultured, the microorganism identification method, the antibiotic treatment and the outcome.

The majority of cases (59.1%) involved newborns that were mostly premature. A slight female predominance was observed (1.2:1). Although *E. anophelis* is ubiquitous in nature with global distribution, most cases (81.8%) have been reported in Asian countries. Meningitis was the most common presentation in newborns. Other clinical manifestations included bloodstream and respiratory infections. The present case was a CLABSI. It has been demonstrated that *E. anophelis* has the ability to form biofilms that facilitate its establishment in CVCs, complicating treatment [23]. The source of the infection and the route of transmission remain unclear for all cases, except for one of vertical transmission from a mother with chorioamnionitis to the neonate [14]. The majority of children had their immune system weakened by prematurity, by intensive medical interventions, or by other comorbidities. The case fatality rate of the infected children was 33.3%, with deaths being most common among infected neonates. Five children among the survivors of *E. anophelis* meningitis developed neurologic sequelae such as hyrocephalus and hearing loss [13,15,18,22].

Accurate identification is essential for selecting the appropriate antimicrobial therapy because of the varying susceptibility profiles among species [24]. It has been shown that inappropriate empirical therapy is an independent risk factor for increased mortality in patients infected with *E. anophelis* [1]. In half of the isolates, the identification of the microorganism was conducted using MALDI-TOF and in the other half by molecular methods or by a combination of MALDI-TOF and molecular methods. The identification of our isolate was performed by MALDI-TOF (v. 3.2), containing, in its database, three species of the genus *Elizabethkingia*, namely *E. meningoseptica*, *E. anophelis* and *E. miricola,* and was further confirmed by 16S rRNA gene sequencing.

*E. anophelis* has been known to be resistant to multiple antimicrobial agents, including most β-lactams, β-lactam/β-lactamase inhibitors, carbapenems and polymyxins [1]. Whole-genome studies revealed numerous antimicrobial resistance-associated genes conferring resistance to β-lactams (such as *bla*_CME-1_, *bla*_blaB_, *bla*_GOB-4,_ and *bla*_CME_)_,_ aminoglycosides, tetracycline, vancomycin, chloramphenicol and multidrug resistance pumps [1]. Several studies have demonstrated conflicting antimicrobial susceptibility testing (AST) results for certain β-lactams, ciprofloxacin, levofloxacin, trimethoprim/sulfamethoxazole and vancomycin [1]. The variations in susceptibility patterns can be attributed to different testing methods. According to Clinical and Laboratory Standards Institute (CLSI) guidelines, the reference antimicrobial susceptibility methods recommended are broth and agar dilution methods [25]. Chiu et al., examining the concordance of AST results, obtained through the gradient diffusion method, with those from the agar dilution method, found agreement between the two methods for ceftazidime, minocycline, doxycycline, levofloxacin and rifampicin [26]. A recent study comparing the broth microdilution susceptibility results of 18 antibiotics against *E. anophelis* with those obtained by the Vitek 2 system found very major discrepancy rates (>1.5%) for ciprofloxacin and moxifloxacin and major discrepancy rates (>3%) for amikacin, piperacillin/tazobactam, tigecycline and trimethoprim/sulfamethoxazole [27]. Of the 12 cases reporting the method used to study antibiotic susceptibility, 7 used the gradient strip method and 5 used automated systems (Phoenix, Vitek 2).

The majority of the reported cases were treated with vancomycin combined with other antibiotics, such as rifampicin, ciprofloxacin, trimthoprim/sulfamethoxazole or piperacillin/tazobactam. The present isolate was resistant to β-lactams, carbapenems and the novel β-lactam/β-lactamase inhibitors such as ceftazidime/avibactam, imipenem/ relebactam and meropenem/vaborbactam because inhibitors have low activity against the metallo-β-lactamases produced by *E. anophelis*. In our case, piperacillin/tazobactam was initially given as empiric therapy and was continued with the addition of amikacin after susceptibility data became available. Notably, four of the nine reported cases of *E. anophelis* meningitis with a favorable outcome were treated with combinations including piperacillin/tazobactam [13,14,17]. Comparable results suggestive of susceptibility to piperacillin/tazobactam have been reported in studies of Comba et al., Jian et al. and Perrin et al. [6,28,29]. However, further evaluation of in vivo data and continuous surveillance of antimicrobial resistance are required to make optimal therapeutic decisions.

## 4. Conclusions

The increasing number of cases of *E. anophelis* infections, which is a result of the availability of new, accurate identification methods, highlights the clinical significance of this opportunistic pathogen in the pathogenesis of human infections. Considering that *E. anophelis* is multiresistant to antibiotics and that inappropriate antimicrobial therapy is an independent risk factor for mortality, the early, accurate identification of bacterial species and prompt effective treatment are essential to achieve optimal therapeutic outcomes.

## Figures and Tables

**Table 1 microorganisms-12-01145-t001:** MICs of isolated *Elizabethkingia anophelis* as determined by gradient strip method.

Antimicrobial Agents	* Breakpoints (μg/mL)			MIC (μg/mL)	Interpretation
	S	I	R		
Piperacillin	≤16	32–64	≥128	≥256	R
Piperacillin/tazobactam	≤16/4	32/4–64/4	≥128/4	12	S
Ceftazidime	≤8	16	≥32	≥256	R
Ceftriaxone	≤8	16–32	≥64	64	R
Cefepime	≤8	16	≥32	16	I
Imipenem	≤4	8	≥16	≥32	R
Meropenem	≤4	8	≥16	≥32	R
Ceftazidime/avibactam	≤8/4	-	≥16/4	12	R
Imipenem/relebactam	≤1/4	2/4	≥4/4	≥32	R
Meropenem/vaborbactam	≤4/8	8/8	≥16/8	≥64	R
Gentamicin	≤4	8	≥16	6	I
Amikacin	≤16	32	≥64	12	S
Plazomicin	≤2	4	≥8	64	R
Tetracycline	≤4	8	≥16	48	R
Doxycycline	≤4	8	≥16	3	S
Minocycline	≤4	8	≥16	0.75	S
Eravacycline	≤0.5	-	>0.5	0.75	R
Tigecycline	≤2	4	≥8	0.75	S
Ciprofloxacin	≤1	2	≥4	0.25	S
Levofloxacin	≤2	4	≥8	0.25	S
TMP/SXT	≤2/38	-	≥4/76	0.19	S
Vancomycin	≤4	8–16	≥32	12	I
Rifampicin	≤1	2	≥4	0.5	S

S, susceptible; I, intermediate; R, resistant; MIC, minimum inhibitory concentration; TMP/SMX, trimethoprim–sulfamethaxole. * CLSI breakpoints for “other non-Enterobacterales” were applied per CLSI document M100-Ed32 guidelines. The breakpoints used for ceftazidime/avibactam, imipenem/relebactam, meropenem/vaborbactam and plazomicin were those reported for Enterobacterales. The breakpoints used for vancomycin and rifampicin were those reported for *Staphylococcus* spp. For tigecycline, the FDA-recommended MIC breakpoints were applied.

**Table 2 microorganisms-12-01145-t002:** Characteristics of pediatric patients with *Elizabethkingiaanophelis* infections.

Ref.	Country of Origin	Age	Sex	Diagnosis	Underlying Conditions	Specimen Type	Identification Method	Antibiotic Treatment	Outcome
[10]	Central African Republic	8 d	F *	Meningitis	Asphyxia at birth	CSF	16S rRNA sequencing	Gentamicin, ampicillin	Death
[13]	China	22 d	M	Meningitis	Prematurity	Blood, CSF	mNGS	Vancomycin, piperacillin/tazobactam	Survival (hydrocephalus)
[13]	China	18 d	F	Meningitis	None	CSF	mNGS	Vancomycin, piperacillin/tazobactam	Survival (hydrocephalus)
[14]	Hong Kong	21 d	M	Meningitis	None	Blood, CSF	rWGS	Vancomycin, piperacillin, rifampicin	Survival (without neurologic sequelae)
[14]	Hong Kong	1 d	F	Meningitis	Prematurity	Blood, CSF	rWGS	Vancomycin, piperacillin/tazobactam, rifampicin	Survival (without neurologic sequelae)
[8]	Hong Kong	1 mo	F	Catheter-related bacteremia	Prematurity, RDS, PDA	Blood	NR	Vancomycin, cefoperazone/sulbactam	Death
[8]	Hong Kong	8 d	F	Meningitis	Imperforated anus, rectovaginal fistula	Blood, CSF	NR	Vancomycin, rifampicin	Survival
[15]	Cambodia	1 d	M	Sepsis	Prematurity	Blood	MALDI-TOF	Imipenem	Survival
[15]	Cambodia	51 d	F	VAP	Ventricular septal defect	Respiratory secretion	MALDI-TOF	Ciprofloxacin	Death
[15]	Cambodia	1 d	M	Sepsis	Prematurity	Blood	MALDI-TOF	Ampicillin, gentamicin	Death
[15]	Cambodia	15 wk	F	Meningitis	Failure to thrive	Blood	MALDI-TOF	Ceftriaxone	Unknown
[15]	Cambodia	8 mo	M	VAP	Duodenal atresia	Respiratory secretion	MALDI-TOF	Meropenem	Death
[15]	Cambodia	7 d	F	Meningitis	Prematurity	Blood	MALDI-TOF	Ciprofloxacin, vancomycin	Survival (hydrocephalus)
[15]	Thailand	1 d	F	Sepsis	Prematurity	Blood	MALDI-TOF	Ampicillin, gentamicin	Death
[16]	India	2 y	F	Bronchopneumonia	NR	Blood	16S rRNA sequencing	Pipercillin/tazobactam, levofloxacin, colistin, ceftriaxone/sulbactam, imipenem	Survival
[17]	India	11 d	M	Meningitis, sepsis	Prematurity	Blood, CSF	16S rRNA sequencing	Pipercillin/tazobactam, vancomycin, ciprofloxacin	Survival (without neurologic sequelae)
[18]	India	12 d	M	Meningitis, sepsis	Prematurity	Blood, CSF	MALDI-TOF	Cefoperazone/sulbactam, vancomycin, TMP/SMX, rifampicin, ciprofloxacin	Survival (hydrocephalus)
[19]	India	7 mo	M	Bacteremia	NR	Blood	MALDI-TOF	Vancomycin, piperacillin/tazobactam	Survival
[20]	Turkey	11 y	M	Bacteremia	Congenital tracheomalacia, cerebral palsy, SARS-CoV-2 past infection	Blood	MALDI-TOF, 16S rRNA sequencing	Colistin, ciprofloxacin	Death
[21]	New York	17 mo	F	Sepsis, pneumonia	None	Blood	WGS	Ampicillin, ceftriaxone, amoxicillin/clavulanate	Survival
[22]	South Carolina	11 d	M	Meningitis, bacteremia	Prematurity	Blood, CSF	MALDI-TOF	Vancomycin, rifampicin, ciprofloxacin, TMP/SMX	Survival (hearing loss, hydrocephalus)
Present case	Greece	2.5 y	F	CLABSI	ALL	Blood	MALDI-TOF, 16S rRNA sequencing	pipercillin/tazobactam, amikacin	Survival

* F, female; M, male; d, days; wk, weeks; mo, months; y, years; VAP, ventilator-associated pneumonia; RDS, respiratory distress syndrome; PDA, patent ductus arteriosus; NR, not reported; CLABSI, central-line associated bloodstream infection; ALL, acute lymphoblastic leukemia; NR, not reported; mNGS, metagenomics next-generation sequencing; rWGS, rapid whole-genome sequencing; WGS, whole-genome sequencing.

## Data Availability

Data are contained within the article.

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
