# Peer review of "Central Line-Associated Bloodstream Infection Due to *Elizabethkingia anophelis*: Case Report and Literature Review on Pediatric Infections"

_microorganisms, 2024, doi:10.3390/microorganisms12061145_

Round 1

Reviewer 1 Report

Comments and Suggestions for Authors

This is a well written case report of Elizabethkingia anophelis CLABSI in a pediatric patient. I did not find anything major which needs revision.

Author Response

We thank the reviewer for the positive comments

Reviewer 2 Report

Comments and Suggestions for Authors

Dear authors,

The subject of the manuscript is appreciated because it is clear that Elizabethkingia spp. is/can become an emerging (nosocomial) pathogen in special patient populations. Therefore, it is extremely important that adherence to diagnostic and therapeutic guidelines is guaranteed.

I would like you to review:

Abstract: rewrite the abstract, taking into account the key messages: difficult identification, difficult AST, difficult (empiric) treatment. 

1. introduction: outline the taxonomic chronology better: Flavobacterium (1959), Chryseobacterium (1994), Elizabethkingia (2005), new species (2011,...)

2. Case description: according to the guidelines the technical diagnosis of CLABSI is executed taking paired blood cultures peripherally and from the CVC. Explain deviation from the guideline. MALDI identification was performed with on plate or with full tube extraction? Are we to expect any differences between the two techniques? Susceptibility testing was executed using E-tests: please use the generic name gradient strip and then refer to the commercial brand (E-test, bioMérieux). Please comment on possible differences with the gold standard method, broth dilution.

Table 1: Please rearrange the columns for better reading, repeat the column headings between the pages intersection. When reading gradient strips, recalculate the MIC result to the nearest higher twofold dilution, ex. 3 mg/L becomes 4 mg/L,...

Comment on the choice of amikacin, with a MIC value of 16 mg/L on the breakpoint, as adjuvant antibiotic.

According to the guidelines, in CLABSI with gram negatives, the CVC is removed. Please comment on the preservation of the CVC: relative thrombobcytopenia, difficult access,...

3. Discussion

Please comment on the soundness of the CLABSI diagnosis, without TTP difference and/or CVC culture, only a possible relation with CVC reservoir can be proven (IDSA/CDC guideline)

Without any genomic based technique (16S, NGS)  species identifcation with proteomics (MALDI) is hazardous. Even in studies where the MALDI database was amended with in house spectra/libraries, the validation had to be very extensive. Please consider genomic confirmation of the identification of the patient strain.

4. Conclusion: please rewrite and focus on what the case has taught you?

5. References:  read and add 

Comba IY, Schuetz AN, Misra A, Friedman DZP, Stevens R, Patel R, Lancaster ZD, Shah A. Antimicrobial Susceptibility of Elizabethkingia Species: Report from a Reference Laboratory. J Clin Microbiol. 2022 Jun 15;60(6):e0254121. doi: 10.1128/jcm.02541-21.

Cheng YH, Perng CL, Jian MJ, Cheng YH, Lee SY, Sun JR, Shang HS. Multicentre study evaluating matrix-assisted laser desorption ionization-time of flight mass spectrometry for identification of clinically isolated Elizabethkingia species and analysis of antimicrobial susceptibility. Clin Microbiol Infect. 2019 Mar;25(3):340-345. doi: 10.1016/j.cmi.2018.04.015.

Comments on the Quality of English Language

Abstract: needs English language editing

2. case description: do not refer to to patient as 'she', replace with 'the patient', 'the case', ...

line 47: A CVC was inserted and the patient was started on intensive chemotherapy

line 71: piperacillin/tazobactam

line 119-120: vertical transmission from a mother with chorioamnionitis to the neonate

line 143-144: please rewrite the sentence. The meaning of it is not clear to me.

Author Response

REVIEWER 2

Abstract: rewrite the abstract, taking into account the key messages: difficult identification, difficult AST, difficult (empiric) treatment. 

Response: We rewrote the abstract emphasizing the key messages of the article.

  1. introduction: outline the taxonomic chronology better: Flavobacterium (1959), Chryseobacterium (1994), Elizabethkingia (2005), new species (2011,...)

Response: In the revised manuscript we have described the taxonomic chronology of the genus.(Lines 41-44)

  1. Case description: according to the guidelines the technical diagnosis of CLABSI is executed taking paired blood cultures peripherally and from the CVC. Explain deviation from the guideline.

Response: In order to establish the diagnosis of CLABSI, paired blood cultures were taken from both the CVC and the peripheral veins  and the differential time to positivity (DTP) was measured. Both sets were positive for the same microorganism (Elizabethkingia anophelis) and the set obtained through the catheter became positive 210 min earlier. This point was  now clarified in the revised manuscript. (Lines 76-77 and 86-87).

  • MALDI identification was performed with on plate or with full tube extraction? Are we to expect any differences between the two techniques?

Response: MALDI identification was performed using direct colony transfer method. A single colony was directly smeared onto the target plate and then overlaid with 1 μl α-cyano-4-hydroxycinnamic acid (CHCA) matrix solution  After air drying at room temperature, the sample was sequentially analyzed by using MALDI-TOF MS (BioMerieux). Laser dependent ionization of bacterial peptides and proteins generated species-specific profiles of spectra allowing bacterial identification by comparison with database profiles of well characterized reference strains.

An article of Wang J et al. evaluating the three sample preparation methods for identification of clinical strains by MALDI-TOF, found that the identification rates were comparable among the three preparation methods but the on-target extraction method (OTEM) is more suitable and necessary for clinical application, owing to its key advantages of simplicity and accuracy.

Wang J, Wang H, Cai K, Yu P, Liu Y, Zhao G, Chen R, Xu R, Yu M. Evaluation of three sample preparation methods for the identification of clinical strains by using two MALDI-TOF MS systems. J Mass Spectrom. 2021 Feb;56(2):e4696. doi: 10.1002/jms.4696.

  • Susceptibility testing was executed using E-tests: please use the generic name gradient strip and then refer to the commercial brand (E-test, bioMérieux).

Response: In the revised manuscript the generic name “gradient strip method” was used and the commercial brand (E-test, bioMérieux), as suggested.

  • Please comment on possible differences with the gold standard method, broth dilution.

Response: According to Clinical and Laboratory Standards Institute (CLSI) guidelines the reference antimicrobial susceptibility methods recommended are broth and agar dilution methods. Chiu et al. examining the concordance of AST results obtained through the gradient diffusion method with those from the agar dilution method, found agreemet between the two methods for ceftazidime, minocycline, doxycycline, levofloxacin and rifampicin. (lines 171-176)

Chiu CT, Lai CH, Huang YH, Yang CH, Lin JN. Comparative Analysis of Gradient Diffusion and Disk Diffusion with Agar Dilution for Susceptibility Testing of Elizabethkingia anophelis. Antibiotics (Basel). 2021;10(4):450.

We also commented on the concordance of microdilution AST results with those obtained by VITEK 2 system. (Lines 176-180).

  • Table 1: Please rearrange the columns for better reading, repeat the column headings between the pages intersection. When reading gradient strips, recalculate the MIC result to the nearest higher twofold dilution, ex. 3 mg/L becomes 4 mg/L,..

Response: Minor changes have been made to Table 1. and  we believe that in the present form it can be easily read. (MIC results remained unchanged).

  • Comment on the choice of amikacin, with a MIC value of 16 mg/L on the breakpoint, as adjuvant antibiotic.

Response: The rationale for giving amikacin as adjuvant antibiotic in combination with piperacillin/tazobactam was that the isolate was susceptible to both antibiotics and their combination acts synergistically.

  • According to the guidelines, in CLABSI with gram negatives, the CVC is removed. Please comment on the preservation of the CVC: relative thrombobcytopenia, difficult access,...

Response: The patient was with a diagnosis of an haematological malignancy post intensive chemotherapy with very low white blood cells, low platelet counts, in a good clinical condition, with no clinical deterioration and improved infection markers. As a result CVC was preserved.

  1. Discussion

Please comment on the soundness of the CLABSI diagnosis, without TTP difference and/or CVC culture, only a possible relation with CVC reservoir can be proven (IDSA/CDC guideline).

Response: In order to establish the diagnosis of CLABSI, paired blood cultures were taken from both the CVC and the peripheral veins  and the differential time to positivity (DTP) was measured. Both sets were positive for the same microorganism (Elizabethkingia anophelis) and the set obtained through the catheter became positive 210 min earlier. This point was  now clarified in the revised manuscript. (Lines 76-77 and 86-87).

Without any genomic based technique (16S, NGS)  species identifcation with proteomics (MALDI) is hazardous. Even in studies where the MALDI database was amended with in house spectra/libraries, the validation had to be very extensive. Please consider genomic confirmation of the identification of the patient strain.

Response: We clarified in the manuscript that the MALDI-TOF database used is validated for clinical application for Elizabethkingia anophelis.

Elizabethkingia anophelis was identified by Matrix-assisted laser desorption ionization-time of flight (MALDI-TOF) mass spectrometry (VITEK MS  system, BioMérieux; version 3.2), containing in its database three species of the genus Elizabethkingia, namely E. meningo-septica, E. anophelis and E. miricola. (Discussion section, lines 159-162).

We did not use MALDI with amended datadase with in house spectra/libraries. Consequently, confirmation of the identification is not necessary. Ιt is no coincidence that 10 of the 20 strains reported in our article were identified only by the use of  MALDI-TOF. Nevertheless, we proceeded to molecular identification by 16S rRNA gene sequencing. The alignment with the Genbank database showed 100% identity with E. anophelis isolate CP034594.1. When the final accession number will be given by the GenBank, will be added to the manuscript. This is expected to happen within the next week.

  1. Conclusion: please rewrite and focus on what the case has taught you?

Response: Conclusion has been rewritten, as suggested by the reviewer.

  1. References:  read and add 

Comba IY, Schuetz AN, Misra A, Friedman DZP, Stevens R, Patel R, Lancaster ZD, Shah A. Antimicrobial Susceptibility of Elizabethkingia Species: Report from a Reference Laboratory. J Clin Microbiol. 2022 Jun 15;60(6):e0254121. doi: 10.1128/jcm.02541-21.

Cheng YH, Perng CL, Jian MJ, Cheng YH, Lee SY, Sun JR, Shang HS. Multicentre study evaluating matrix-assisted laser desorption ionization-time of flight mass spectrometry for identification of clinically isolated Elizabethkingia species and analysis of antimicrobial susceptibility. Clin Microbiol Infect. 2019 Mar;25(3):340-345. doi: 10.1016/j.cmi.2018.04.015.

Response: The above cited references are very important and have been added in the revised manuscript, as suggested by the reviewer.

Abstract: needs English language editing

Response: English language has been corrected in the abstract, as recommended by the reviewer.

case description: do not refer to to patient as 'she', replace with 'the patient', 'the case',

Response: We have replaced “she” with the “patient”

line 47: A CVC was inserted and the patient was started on intensive chemotherapy

Response: The phrase was rewritten, as suggested.

line 71: piperacillin/tazobactam

Response: “piperacillin/tazobactam” was corrected.

line 119-120: vertical transmission from a mother with chorioamnionitis to the neonate

Response: The phrase was corrected.

line 143-144: please rewrite the sentence. The meaning of it is not clear to me.

Response: The sentence was rewritten.

Thank you very much.

Reviewer 3 Report

Comments and Suggestions for Authors

Excellent paper! All the necessary aspects have been covered and discussed in sufficient details. The case is presented nicely (my only comment would be that it was only later mentioned where the case happened- later I saw Greece in Table 2), and the review of literature and previously published case reports is provided with all essential infos.

Myself, as reader, would have liked if any more data were given on diagnostics of bacteria in other cases-however I am aware that this is not the mandatory part for this type of article.

The only thing I would highly request- lines 70-75- it is not necessary to list antibiotic, as all the data are given in Table 1. You can simply say- The susceptibility to antibiotics is given in Table 1, or something along the line.

Please make sure that you correct the spelling mistakes (found several of them), and to define all the abbreviations.

Best of wishes in publishing your paper! 

Author Response

REVIEWER 3

Excellent paper! All the necessary aspects have been covered and discussed in sufficient details. The case is presented nicely (my only comment would be that it was only later mentioned where the case happened- later I saw Greece in Table 2), and the review of literature and previously published case reports is provided with all essential infos.

Myself, as reader, would have liked if any more data were given on diagnostics of bacteria in other cases-however I am aware that this is not the mandatory part for this type of article.

Response: Thank you for the comment. In the revised manuscript,we added in Table 2 a column containing data regarding the microorganism’s identification method in all cases.

The only thing I would highly request- lines 70-75- it is not necessary to list antibiotic, as all the data are given in Table 1. You can simply say- The susceptibility to antibiotics is given in Table 1, or something along the line.

Response: We agree with the reviewer. We omitted the list of antibiotics and added the phrase “The antimicrobial susceptibility pattern of the isolate is presented in Table 1.

Please make sure that you correct the spelling mistakes (found several of them), and to define all the abbreviations.

Response: Spelling mistakes were corrected and all the abbreviations were defined.

Best of wishes in publishing your paper! 

Thank you very much.

Round 2

Reviewer 2 Report

Comments and Suggestions for Authors

Dear authors,

Thank you for revising the manuscript, taking into account the suggestions. I consider the sequencing results of the microorganism a very major improvement.

I'm still not sure that you understood correctly my comments on the  limitations of MALDI identification. It doesn't show clearly in the text.

Paragraph line 103-110 has to be rewritten: Amikacin was added as an adjuvant antibiotic because of the synergistic (or complementary?) action that can be expected with piperacillin-tazobactam. Four days after the beginning of the episode the patient became afebrile and blood cultures remained negative. Despite the initially very low white blood cell and platelet counts post intensive chemotherapy, clinical and biochemical improvement was observed. Intravenous antibiotics were continued for 10 days and the CVC was preserved. Currently, the patient receives maintenance treatment with oral chemotherapeutical agents.

line 132-34: ...summarizes the patient characteristics, the clinical manifestations.....the microorganism identification...

line 175 agreement

Comments on the Quality of English Language

See previous comments.

The English language remains of average quality but no real issues are detected.

Author Response

Thank you for revising the manuscript, taking into account the suggestions. I consider the sequencing results of the microorganism a very major improvement.

I'm still not sure that you understood correctly my comments on the  limitations of MALDI identification. It doesn't show clearly in the text.

Response: We clarified the limitations of MALDI-TOF using the reviewer’s own words.(Lines 123-125)

Paragraph line 103-110 has to be rewritten: Amikacin was added as an adjuvant antibiotic because of the synergistic (or complementary?) action that can be expected with piperacillin-tazobactam. Four days after the beginning of the episode the patient became afebrile and blood cultures remained negative. Despite the initially very low white blood cell and platelet counts post intensive chemotherapy, clinical and biochemical improvement was observed. Intravenous antibiotics were continued for 10 days and the CVC was preserved. Currently, the patient receives maintenance treatment with oral chemotherapeutical agents.

Response: The paragraph line 103-110 has been rewritten, as per reviewer’s suggestion. .(Lines 102-108)

line 132-34: ...summarizes the patient characteristics, the clinical manifestations.....the microorganism identification...

Response: The sentence was changed, as suggested by the reviewer. (Lines 131-132)

line 175 agreement

Response: “agreemet” was corrected to “agreement”